# BinVPR: Binary Neural Networks towards Real-Valued for Visual Place Recognition

**DOI:** 10.3390/s24134130

**Published:** 2024-06-25

**Authors:** Junshuai Wang, Junyu Han, Ruifang Dong, Jiangming Kan

**Affiliations:** 1School of Technology, Beijing Forestry University, Beijing 100083, China; junshuai1043@bjfu.edu.cn (J.W.); hanjunyu0801@bjfu.edu.cn (J.H.); kanjm@bjfu.edu.cn (J.K.); 2Key Laboratory of State Forestry Administration on Forestry Equipment and Automation, Beijing 100083, China

**Keywords:** visual place recognition, binary neural networks, model compression, gradient vanishing, gradient mismatch

## Abstract

Visual Place Recognition (VPR) aims to determine whether a robot or visual navigation system locates in a previously visited place using visual information. It is an essential technology and challenging problem in computer vision and robotic communities. Recently, numerous works have demonstrated that the performance of Convolutional Neural Network (CNN)-based VPR is superior to that of traditional methods. However, with a huge number of parameters, large memory storage is necessary for these CNN models. It is a great challenge for mobile robot platforms equipped with limited resources. Fortunately, Binary Neural Networks (BNNs) can reduce memory consumption by converting weights and activation values from 32-bit into 1-bit. But current BNNs always suffer from gradients vanishing and a marked drop in accuracy. Therefore, this work proposed a BinVPR model to handle this issue. The solution is twofold. Firstly, a feature restoration strategy was explored to add features into the latter convolutional layers to further solve the gradient-vanishing problem during the training process. Moreover, we identified two principles to address gradient vanishing: restoring basic features and restoring basic features from higher to lower layers. Secondly, considering the marked drop in accuracy results from gradient mismatch during backpropagation, this work optimized the combination of binarized activation and binarized weight functions in the Larq framework, and the best combination was obtained. The performance of BinVPR was validated on public datasets. The experimental results show that it outperforms state-of-the-art BNN-based approaches and full-precision networks of AlexNet and ResNet in terms of both recognition accuracy and model size. It is worth mentioning that BinVPR achieves the same accuracy with only 1% and 4.6% model sizes of AlexNet and ResNet.

## 1. Introduction

Visual Place Recognition (VPR) uses visual information to determine whether a robot or autonomous agent has previously visited a place [1]. VPR is an essential technology for robot localization and navigation [2]. It enables a robot to localize itself and correct the incremental drift of its pose estimation during navigation. VPR can be considered as an image retrieval system that, given a query image, retrieves the most similar locations in the stored database. In recent years, VPR approaches based on Convolutional Neural Networks (CNNs) have attracted the attention of numerous researchers, which are more robust and discriminative than those based on hand-crafted VPR approaches.

The success of CNN models depends dramatically on their high computational cost and heavy parameters. However, it is a great challenge for mobile robot platforms equipped with limited resources [3]. Therefore, reducing the model size is a crucial way to make it applicable to mobile robots. Ferrarini et al. [4] proposed FloppyNet that used a Binary Neural Network (BNN) to reduce the model size while representing an extreme scenario of model quantization by using 1-bit instead of 32-bit floating for weighting and activation. AlexNet was binarized in [4]. Its network depth was also reduced, the model size was decreased to 99% of AlexNet, and computation efficiency was significantly increased by seven times.

However, current BNNs always suffer from gradient vanishing in the training process and a marked drop in accuracy, which results from the conversion of 32-bit values into 1-bit values, causing severe loss of information. Thus, the BNN-based VPR approaches require further investigation. Moreover, inspired by FloppyNet, this work constructed a baseline network based on ResNet [5]. ResNet was selected due to its fewer parameters and better accuracy than AlexNet.

For the gradient-vanishing problem, we proposed the following hypothesis: In the high-level structure of the binary network, too much information was lost, and the gradient could not be effectively accumulated during backpropagation, which caused the gradient-vanishing phenomenon. This hypothesis was inspired by MobiNet [6] and verified by designing experiments. Based on this, we further proposed a feature restoration strategy to reintroduce part of the feature information into the subsequent convolutional layers. This paper explores which features should be restored and how the network should accept these features to minimize the gradient-vanishing problem. We divided the added features into three categories according to their source positions: basic, intra-block, and inter-block features. This work explored the effect of all added features, and the best one was obtained. Meanwhile, an exploration of the optimal position to accept these restoration features was also conducted. Finally, we concluded with two key principles for designing a BNN architecture that can effectively deal with the gradient-vanishing problem.

For the marked drop in accuracy of BNNs, Ding et al. [7] indicated that it results from the gradient mismatch problem, which usually occurs in backward propagation. To weaken the gradient mismatch problem, an efficient way is to optimize the combination of binarized activation and binarized weight functions. Thus, taking a binarization ResNet-18/34 with a full-precision short-cut connection as an example, we studied all the possible combinations of activation and weight signs in [8,9]. It owes to the brute-force approach. Finally, the optimal combination was found. With the above description, our BinVPR can be proposed. It achieved the same level of accuracy as the full-precision network but with a significantly reduced model size. The contributions of this paper are as follows:(1)A feature restoration strategy was proposed to solve the gradient-vanishing problem, which aims to supply some lost information to higher levels of the network. In order to restore the lost information, we explored different kinds of features to add different positions to accept these features. Thereby, the best one was obtained.(2)Two principles have been found that can solve the problem of vanishing gradients in the training of BNNs and determine the structure of BNNs. These principles include restoring basic features to solve the gradient-vanishing problem and restoring basic features from the high layer to the low layer by layer.(3)A brute-force approach was introduced to find the optimal combination of binarized activation and binarized weight function to further improve the dropped accuracy caused by the gradient mismatch.(4)A baseline network based on ResNet was designed, and the proposed binary network BinVPR was presented to realize visual place recognition. The performance of BinVPR was tested on VPR public datasets. The results show that BinVPR outperforms state-of-the-art CNN-based approaches such as AlexNet and ResNet in terms of both accuracy and model size.

This paper is organized as follows. The next section reviews relevant literature, while Section 3 presents a step-by-step process for obtaining concise BNNs for VPR. Section 4 proposes a comprehensive analysis of binary layers for VPR applications. Finally, Section 5 concludes this article.

## 2. Related Work

### 2.1. Deep Learning for VPR

Deep Learning (DL) has received considerable attention for its remarkable success in Computer Vision (CV) [10,11,12] and Robotics [13], mainly due to the widespread use of Convolutional Neural Networks (CNNs). Pre-trained CNN models are usually used as feature extractors. Sünderhauf et al. [14] and Nasser et al. [15] used AlexNet pre-trained on the ImageNet dataset. PlaceNet [16] is based on the same principle and is trained on a large dataset called Places365, which is organized into 365 categories. In addition to using pre-existing CNN architectures, specific VPR models are also being developed and trained, such as NetVLAD [17], which replaces the last fully connected layer of the original CNN model to enable end-to-end training for large-scale place recognition. Patch-NetVLAD [18] combined the advantages of local and global descriptor methods to extract patch-level features from NetVLAD residuals. Sun et al. [19] proposed a modified patch-NetVLAD strategy, called contextual Patch-NetVLAD. Contextual Patch-NetVLAD first aggregated features from each patch’s surrounding neighborhood. Then, this method utilized cluster and saliency context-driven weighting rules to assign higher weights to patches. Xin et al. [20] proposed a Landmark Localization Network (LLN) that predicts the discrimination of local features for corresponding regions within an image to generate discriminative landmarks. Izquierdo et al. proposed SALAD [21] (Sinkhorn Algorithm for Locally Aggregated Descriptors). Considering the feature-to-cluster and cluster-to-feature relationships, they introduced the “dustbin” cluster to discard the features that are considered to be non-informative and improve the overall quality of the descriptors. It is worth noting that Ali-bey proposed the GSV-Cities [22] dataset to address the challenge of lacking large databases with accurate ground truth. Ali-bey collected more than 40 cities across all continents over a 14-year period, providing the widest geographic coverage and highly accurate ground truth. To realize seamless adaptation of the pre-trained transformer model for VPR, Lu et al. [23] proposed the SelaVPR model. This model design is a hybrid adaptation that adjusts the lightweight adapter. Moreover, they proposed a mutual nearest neighbor local feature loss to guide effective adaptation, which reduces the matching time of the image.

### 2.2. Binary Neural Network

Convolutional Neural Networks (CNNs) exhibit strong learning ability owing to their multi-layered structure and millions of parameters. This results in a substantial size and significant computational burden. Therefore, a vital issue for VPR is to reduce the model size. Binary neural networks can drastically reduce resource demand and improve computational efficiency. BinaryConnect [24] proposed stochastic binarization that is used during forward propagation of the training to quantize weights. This method incorporates a clipping function to cancel the gradient when activation exceeds 1.0, which improves accuracy. Afterward, some works were proposed to improve BNNs. XNOR-Net [25] addresses this accuracy gap by introducing a channel-wise scaling factor, which is obtained by $l_{1}$-norming weights or activations. This reduces the error of full precision. DoReFa-Net [26] investigated neural networks trained with 1-bit weights and 2-bit activations. DoReFa-Net investigated neural networks trained with 1-bit weights and 2-bit activations. Compared to the original AlexNet model, this method’s accuracy decreased by 4.9% on the ImageNet dataset. Bi-Real-Net [27] is proposed on the basis of ResNet [5]. It employs a highly sophisticated training strategy consisting of full-precision pre-training, multi-step initialization, and user-defined gradients, to achieve viability for real-world applications. Moreover, Wang et al. proposed BitNet [28] for a large language model, which is a scalable and stable 1-bit transformer architecture. They designed the trainable 1-bit fully connected layer BitLinear instead of the nn.Linear layer. BitNet achieved competitive performance while reducing memory footprint and energy consumption, compared to full-precision Transformer baselines. Xue et al. [29] proposed ReBNN to improve learning ability. This method reduced the loss of the binarization process by calculating the balanced parameter based on its maximum magnitude.

FloppyNet [4] used AlexNet as a baseline network to construct a binary network for VPR, which improved operational efficiency and saved memory. Inspired by FloppyNet, we constructed a baseline network based on ResNet to overcome the gradient-vanishing problem and the decreased accuracy problem because ResNet has fewer parameters and better performance than AlexNet.

### 2.3. Application of VPR in vSLAM

VSLAM [15] (Visual Simultaneous Localization and Mapping) uses visual information to tasks if it is possible for a mobile robot to be placed at an unknown location in an unknown environment, at the same time, and builds a consistent map of this environment while simultaneously determining its location within this map. VSLAM systems have four main components: visual odometry, Loop Closure Detection (LCD), back-end optimization, and mapping. True loop closure reduces [30] the cumulative position errors caused by visual odometers and builds accurate and consistent maps. The LCD can also be regarded as place recognition [1] and is an important part of VSLAM systems [31]. This problem involves giving a location image and determining whether the place exists in the location database.

At present, the main applications of VPR in vSLAM include VPR based on local feature descriptors, VPR based on global feature descriptors, and VPR based on learning method. VPR based on local feature descriptors requires a detection phase that determines individual patches or key points within the image to retain as local features, including SIFT [32], SURF [33], and ORB [34]. In contrast, VPR based on global feature descriptors such as WI-SURF [35] and BRIEF-Gist [36] does not have a detection phase but process the whole image regardless of its content. In recent years, with the development of deep learning, researchers have tried to use this method to learn the expression of global images from visual information. The learning-based VPR approach is described in Section 2.1.

From the perspective of the specific application of VSLAM, VPR provides loop closure information for the backend. Incorrect loop information leads to drastic degradation of map quality. Therefore, a place recognition algorithm applied to VSLAM should have high accuracy. The advantages of local and global feature descriptors are compact representation and computational efficiency, leading to lower storage consumption and faster indexing when retrieving location images. VPR based on global features descriptors shows strong robustness to illumination, but cannot handle occlusion and incorporate geometric information. VPR based on local feature descriptors is robust to rotation and scale and can be recognized even under partial occlusion. In addition, these methods are incorporated into spatial information and then combined with the metric pose estimation algorithms, which are successfully applied in VSLAM, such as ORB-SLAM [37]. VPR based on learning can combine the advantages of local and global features and has high accuracy in the changing scaling, illumination, and occlusion environment. However, VPR based on learning requires higher computing and storage resources. Therefore, reducing the requirement of deep learning on computing and storage resources has become one of the key issues in the study of VPR.

## 3. Methodology

In this section, the proposed BinVPR model is presented. Firstly, a baseline network based on ResNet’s [5] plain network is constructed. Then, the brute-force approach is described to search for the optimal combination of binarized weights and binarized activation functions, thereby improving the accuracy of the binary network model. Moreover, the feature restoration strategy is interpreted. Basic, intra-block, and inter-block features were added to higher levels of the network to find the most suitable features to restore in order to solve the gradient-vanishing problem. Finally, we introduced the method of designing binary networks to solve the problem of gradient vanishing during the training process for BNNs.

### 3.1. Baseline Network

We designed our baseline network (see in Figure 1) based on the plain network of ResNet-18/34. Through experiments, we determined the final network structure on the baseline network.

The basic block was established by stacking three convolutional layers whose network width is 128. The weights of the basic block are binarized, but the input activation of the basic block is not binarized. This approach helped to retain more information. The top layer of the basic block extracted features as basic features.

Zagoruyko and Komodakis [38] highlighted the ability of wider networks to capture more features and allow for smoother training. Therefore, we increased the network width of the plain network. In the main part of the network (except the basic block in the baseline network), the width of the first block was increased from 64 to 128, the width of the second block was increased from 128 to 256, the width of the third block was increased from 256 to 384, and the width of the fourth block was unchanged.

In the main part of the network, the inputs and weights were binarized. The fully connected layer outputted the feature vector. Thus, our baseline networks, i.e., Baseline-20 and Baseline-36, were constructed as Figure 1 shows.

### 3.2. Original Binarization

Binarization converts floating values to binary values, effectively reducing memory. Courbariaux and Bengio [24] first proposed the binarization function Straight-Through Estimator (STE) coupled with gradient clipping, as introduced by Hubra et al. The weights and activations are binarized using the following sign function:
(1)$$x_{b}=sign\left(x\right)=\left\{\begin{array}{cc} 1, & \mathrm{if}\;x\geq 0,\\ -1, & \mathrm{otherwise}. \end{array}\right.$$
where *x* is a real-valued variable and $x_{b}$ is a binary-valued variable. In forward propagation, STE performs sign functions. In the backward propagation phase, *x* updates according to the network’s loss gradient. Let *l* denote the loss function, $r_{i}$ be a real number input, and $r_{o}\in \left\{-1,+1\right\}$ be a binary output. Furthermore, $t_{clip}$ is a threshold for clipping gradients and was originally set to be 1. The function returns a clipped identity of gradient in the backward phase. Therefore, the final STE formula can be stated as follows:(2)$$\begin{array}{cc} Forward:r_{o} & =sign(r_{i}). \end{array}$$
(3)$$\begin{array}{cc} Backward:\frac{\partial l}{\partial r_{i}} & =\frac{\partial l}{\partial r_{o}}1_{\left|r_{i}\right|\leq t_{clip}}. \end{array}$$

Gradient clipping helps the optimization process of the binary network because backpropagation no longer increases the absolute value of the input more larger than the clipping threshold, similar to regularization in the full-precision network.

### 3.3. Optimization for Combination of Activations and Weights Sign

Because of the limited expression ability of the original binarization function, the recognition performance of the binary network will be greatly reduced. Later, the binarized activation value and weight value used different binarization functions. However, this also presents a problem. As Ding et al. [7] pointed out, different binarization functions may have gradient mismatches during training. To solve this problem, we optimized the combination of binarization activation functions and binarization weight functions.

We conducted comprehensive experiments training on “Forest” to find the best combinations of LARQ’s built-in binarization activation and weight functions. The experimental results show that using “leaky tanh” for binarizing activation combined with DoReFa-Net’s weight function yields the most favorable performance results. In Larq’s framework, they provided the binarized activation function of “leaky tanh”. The sign function is as follows:
(4)$$a_{o}=\left\{\begin{array}{cc} 1+(a_{i}-1)\cdot \alpha , & \mathrm{if}\;a_{i}>1,\\ a_{i}, & \mathrm{if}-1\leq a_{i}\leq 1,\\ -1+(a_{i}+1)\cdot \alpha , & \mathrm{otherwise}. \end{array}\right.$$
where $a_{o}$ represents the output of activation, $a_{i}$ represents the input of activation, and α is 0.2 in Larq.

DoReFa-Net’s binarization weight function used k−bit representation of the weights with k>1 and used the STE $f_{w}^{k}$ to weights as follows:
(5)$$\begin{array}{cc} Forward:r_{o}=f_{w}^{k} & =2Q_{k}\left(\frac{tanh(r_{i})}{2max(|tanh\left(r_{i}\right)|)}+\frac{1}{2}\right)-1. \end{array}$$(6)$$\begin{array}{cc} Backward:\frac{\partial l}{\partial r_{i}} & =\frac{\partial r_{o}}{\partial r_{i}}\frac{\partial l}{\partial r_{o}} \end{array}$$
where $r_{i}$ is the real-value input and $r_{o}$ is the binary output. $Q_{k}$ is quantized as a real-number input $r_{i}\in \left\{0,1\right\}$ to a k−bit-number output $r_{o}\in \left[0,1\right]$. The maximum value is taken over all weights in this layer. The backward sign is as shown in Equation (3), and the forward function is as follows:(7)$$r_{o}=\frac{1}{2^{k}-1}round\left((2^{k}-1\right)r_{i}).$$

It is worth noting that they used the tanh function to constrain the weight values to the range [0, 1] prior to quantization into k−bit. Through this process, $\frac{tanh(w_{i})}{2max(|tanh\left(w_{i}\right)|)}+\frac{1}{2}$ is designed to produce a number in the range [0, 1] that reaches the highest value of all the weights. The range of $f_{w}^{k}\left(w_{i}\right)$ is then adjusted to [−1, 1] by the subsequent affine transform. This approach ensures standardized and bounded weight values for additional computation.

### 3.4. Feature Restoration Strategy

This section presents our three feature restoration strategies, i.e., basic feature restoration strategy, inter-block and intra-block feature restoration strategy.

#### 3.4.1. Basic Feature Restoration Strategy

According to literature [6], the features in higher layers of the network suffer from more severe information loss, which leads to gradient vanishing during the training process. Thus, in the basic feature restoration strategy, basic features extracted from the basic block were added to higher convolutional layers to provide more information. Regarding which convolutional layer accepts basic features, this work gives two ways, as Figure 2 shows, i.e., restoration for a single layer and restoration for multiple layers. In the restoration for a single layer way, basic features were added to a single layer, ranging from the second layer of Block 2 to the last ReLU; thus, there are 16 ways in total. In the restoration for multiple layers way, basic features were supplied to multiple layers at the same time, and the multiple layers to accept basic features are added from the last ReLU, as Figure 2 shows. There are also 16 ways in total. Experiments were performed in Section 4 to compare the effect of restoration for different layers and to find the best one from these 32 ways.

#### 3.4.2. Inter-Block Feature Restoration Strategy

For the inter-block feature restoration strategy, the first convolutional layer of a block was defined as inter-block features. They were fed into the convolutional layer of the other block. As Figure 3 shows, it includes three restoration ways, i.e., the abbreviated “single block—single layer”, “multiple blocks—single layers”, and “multiple blocks—multiple layers”. Here,“single block—single layer” represents the inter-block features of the current block that were added to a single convolutional layer of another block, as the black dotted line shows in Figure 3; “multiple blocks—single layers” means the inter-block features of the current block were simultaneously added to a single convolutional layer at the same location in other blocks, as the blue dotted line shows in Figure 3; and “multiple blocks—multiple layers” means the inter-block features of multiple blocks were supplied to multiple convolutional layers of another block at the same time, as the red dotted line shows. For Baseline-20, there are 4 blocks, leading to 42 restoration ways in total. Experiments were performed in Section 4 to compare the effect of the total 42 ways.

#### 3.4.3. Intra-Block Feature Restoration Strategy

The output of the first convolutional layer of a block was defined as basic features of this block. The intra-block feature restoration strategy extracts the basic features of a block to add into the rest layers of this block, as Figure 4 shows. It includes 2 restoration ways, i.e., the abbreviated “single block” and “multiple blocks”. Here, “single block” represents a restoration way where the basic features of a block were added to one or more convolutional layers in this block, and “multiple blocks” denotes a restoration way where there are multiple blocks performing “single-block” restoration at the same time. For Baseline-20, there are 21 restoration ways in total. Experiments were performed in Section 4 to compare the effect of the total 21 ways.

#### 3.4.4. How Features Are Restored

ResNet applied residual block learning for every few stacked layers. Figure 5 shows 18 and 34 layers of the ResNet building block. The block can be defined as:(8)$$y=F(x,\left\{w_{i}\right\})+x$$
where *x* and *y* are input and output vectors of the considered layers. $F(x,\left\{w_{i}\right\})$ represents the residual map to be learned. They add the latter extraction features after convolutional operations. Unlike short connection, feature restoration supplies features to the latter convolutional layers. We have defined this operation as:(9)$$y=F(x+f\left(x_{s}\right),\left\{w_{i}\right\})$$
where *x* is input vectors and *y* is output vectors. $x_{s}$ represents the features to be restored. Function *F* represents the restoration features map to be learned. The function $f(x_{s})$ represents the number of dimensions aligned between $x_{s}$ and *x*, which uses a full-precision convolutional layer with a kernel size of 1 × 1, and the stride is a multiple of 2. The restoration features operation is shown in Figure 5a–c.

### 3.5. Design Binary Network

For the problem of gradient vanishing encountered in the training process of BNNs, we conducted an in-depth exploration. After analysis, we found that gradient vanishing is mainly due to the limited representation ability of binarization method, which made the accumulated gradient insufficient in the backpropagation process and led to this problem in the training process. Therefore, to address this problem, it needed to restore some of the features in the binary network. However, how should features be restored and what features should be restored so became the central question of our research.

How should features be restored?

Restoring features that retain more information can solve the problem of gradient vanishing. We have conducted in-depth research and found that basic features play a key role in solving the gradient-vanishing problem. In contrast, intra-block and inter-block features failed to alleviate this phenomenon effectively. Even if we add all the intra-block and inter-block features to the high-level network, the problem remains. We further analyzed that these features showed different effects because the basic features were not binarized, but the inter-block and intra-block features were binarized. Binarization features lose a lot of information, so solving the gradient disappearance problem is difficult. Restoring those features that have been un-binarized can effectively solve the phenomenon of gradient vanishing.

What features should be restored?

To solve the gradient-vanishing problem, we adopted a strategy, feature restoration. This strategy is a layer-by-layer feature restoration from the high layer to the low layer of the network until the gradient-vanishing phenomenon is solved. During the experiment, we first tried to recover the basic features from the low-level network, but the effect was insignificant. We then turned to high-level networks for feature recovery and found that this solved the vanishing gradient problem. Because the low-level network had less information loss in the binarization process to accumulate enough gradients in the backpropagation, and it was less apparent in the gradient-vanishing phenomenon. However, with the increase in network layers, the information loss was gradually aggravated, and the gradient accumulated was gradually reduced, resulting in the gradient-vanishing phenomenon becoming increasingly apparent. Therefore, restoring features should start from the high-level network and advance layer by layer to lower layers to solve the gradient-vanishing problem.

## 4. Experiments

This section begins with an introduction to the experimental dataset in Section 4.1. Subsequently, a comprehensive ablation study was conducted in Section 4.2 to thoroughly evaluate the efficacy of the proposed feature restoration method. Following this, we conducted experiments to determine the optimal combination between activation and weight functions to address the challenge of binarizing input activations and weights. Finally, in Section 4.4, we compared our work with other state-of-the-art binary networks and full-precision neural networks regarding accuracy and model size.

### 4.1. Training Dataset

We chose the Places365 [16] dataset as the experimental dataset. Places365 conforms to human visual cognition and can be used to train artificial neural networks for advanced visual tasks. It contains over a million images and has 365 image categories, each with 3000 to 5000 images. VPR faces the challenges of perceptual aliasing and variability in a changing environment. Due to seasonal changes, lighting conditions, and object occlusion, the place’s appearance changes greatly, leading to perceptual aliasing. In addition, when the location structure of the scene is similar, it is easy to cause perceptual variability. These challenges are particularly prominent in complex forest environments, and therefore, we grouped forest scenes in particular. We divided the other scenes into indoor scenes, outdoor human-made scenes, and natural scenes. Torii et al. [39] subdivided scenes into two broad categories based on whether or not the image has a repetitive structure: human-made and natural scene.

Further, human-made scenes are divided into indoor scenes and outdoor human-made scenes. The outdoor human-made scene has many repetitive structures and was greatly affected by natural factors such as light and seasonal changes. In contrast, in indoor scenes, VPR is usually unaffected by light and seasonal changes and has a large number of repetitive structures. The structures in the natural environment are similar but not repetitive and are also affected by natural factors. Therefore, to improve the accuracy of VPR, we subdivide the Places365 dataset into four categories Table 1 and use these data to test and verify BNNs in the experiment.

Specifically, “Forest” was constructed by selecting a few scene classes from Places365, which includes deciduous forest, desert vegetation, rainforest, bamboo forest, and tree farm. “Natural scene” chose 48 classes of natural environment images from Places365, including mountains, rivers, lakes, seas, etc. “Indoor scene” selected 156 classes of indoor images such as bars, classrooms, etc. Meanwhile “Outdoor human-made building” selected 156 categories of images representing outdoor buildings. See Table 1 for more details.

### 4.2. Evaluation Metric

We used Top-1 accuracy in the paper. Top-1 accuracy means the best guess (class with the highest probability) is the correct result. Top-5 accuracy means the correct result is in the top five best guesses (five classes with the highest probabilities). Compared with Top-5 accuracy, Top-1 accuracy is more serious and straightforward. It measures the model’s ability to predict the most likely class correctly. Moreover, in some datasets we trained, the maximum scenarios are 10. Top-5 accuracy cannot provide enough discrimination, and most predictions are in the top five. Therefore, we chose Top-1 accuracy as the evaluation metric.

### 4.3. Evaluation of Feature Restoration Strategy

A series of experiments were performed to assess the effectiveness of different types of feature restoration on “Forest”. The evaluation index is recognition accuracy.

Firstly, in order to evaluate the effectiveness of the proposed basic feature restoration strategy and find the optimal position to add features, the restoration for a single layer and restoration for multiple layers, which include 32 ways in total, were compared. For restoration for a single layer, we added basic features to different places in the network, ranging from the first layer to the last layer. It is discovered that supplying basic features to the last convolutional layer and the last ReLU layer achieved the best results. They can overcome gradient vanishing: the recognition accuracies were 74.75% and 73.25%, respectively. However, supplying basic features to lower layers cannot the solve the gradient-vanishing problem. Therefore, it can be concluded that adding basic features to higher layers is more beneficial than lower layers.

In the restoration for multiple layers way, basic features were supplied to multiple layers at the same time. It aims to conclude which layers are the best choices to accept basic features. Based on the above conclusion, basic features thereby were restored from the last ReLU to a few lower layers for Baseline-20 and Baseline-36. Results are shown in Figure 6, where the *y*-axis shows accuracy. Gradient vanishing appears when accuracy is lower than 0.3. The *x*-axis gives places where basic features were restored. “Zero” represents a “plain network” and without features supplied. The abbreviated “LR.” represents the place that accepts basic features in the last ReLU layer of the whole network. “1st-LC” represents the place that accepts basic features, including the last ReLU layer and the last convolutional layer of the whole network. Similarly, “2nd-LC” represents the place that accepts basic features, including the last ReLU layer and the last two convolutional layers of the whole network, and so forth. Figure 6 indicates that Baseline-20 can solve the gradient-vanishing problem when basic features are supplied to place “LR.” and after that. Moreover, when basic features were supplied to place “1st-LC”, it achieved the best accuracy, nearly the same value as ResNet-18. In addition, the results of Baseline-36 show that it can solve the gradient-vanishing problem when basic features are supplied to place “6th-LC” and after that. When basic features were supplied to place “7th-LC”, it achieved the best accuracy, nearly the same value as ResNet-34.

It can be therefore concluded that basic feature restoration is able to solve the gradient-vanishing problem. Meanwhile, the deeper the whole network, the higher the convolutional layers need to be to restore basic features after binarization of the full-precision model. Moreover, basic feature restoration is beneficial to improve the lost accuracy.

Secondly, the proposed inter-block and intra-block feature restoration strategies were also evaluated. For inter-block feature restoration, a total of 42 restoration ways were compared, and the results are shown in Table 2. The first three parts of “Block 2”, “Block 3”, and “Block 4” belong to the inter-block feature restoration of “single block—single layer”. For example, in the “Block 2” part, the location of row “2nd” and column “1st B1” indicates that the inter-features of Block 1 were supplied to the second convolutional layer of Block 2. Other representations are similar. The middle parts of “Block 2 + 3”, “Block 3 + 4”, and “Block 2 + 3 + 4” belong to the inter-block feature restoration of “multiple blocks - single layers”. For example, in the “Block2+3” part, the location of row “2nd” and column “1st B1” indicates that the inter-features of Block 1 were supplied to the second convolutional layer of Blocks 2 and 3. Other representations are similar. The last three parts of “Block2+3”, “Block3+4”, and “Block 2 + 3 + 4” belong to the inter-block feature restoration of “multiple blocks - multiple layers”. For example, in the “Block 2 + 3” part, the location of row “2nd + 3rd” and column “1st B1” indicates that the inter-features of Block 1 were supplied to the second and third convolutional layers of Block 2 and 3 at the same time. Other representations are similar. From this table, we can see that inter-block feature restoration cannot solve the gradient-vanishing problem.

For intra-block feature restoration, as Section 3 described, there are 21 restoration ways in total. The experimental results are shown in Table 3. In the table, the rows from the “1st layer of B1” to the “1st layer of B4” belong to the intra-block feature restoration of “single block”. For example, the location of row “1st layer of B1” and column “2nd” indicates that basic features of Block 1 were supplied to the second convolutional layer of Block 1, while the location of row “1st layer of B1” and column “2nd + 3rd” indicates that basic features of Block 1 were supplied to the second and third convolutional layers of Block 1, etc. The rows from “1st layer of B1+2” to “1st layer of B1+2+3+4” belong to the intra-block feature restoration of “multiple blocks”. For example, the location of row “1st layer of B1+2” and column “2nd” indicates that basic features of Block 1 were supplied to the second convolutional layer of Block 1, at the same time, basic features of Block 2 were supplied to the second convolutional layer of Block 2. As another example, the location of row “1st layer of B1+2” and column “2nd + 3rd” indicates that basic features of Block 1 were supplied to the second and third convolutional layers of Block 1 at the same time; meanwhile, basic features of Block 2 were supplied to the second and third convolutional layers of Block 2. We can see that intra-block feature restoration cannot solve the gradient-vanishing problem. BNNs will have a more severe gradient disappearance problem as the network depth increases. Therefore, in inter-block and intra-block feature restoration, the features supplied from high layers are futile.

Finally, we tested the effect of combining basic feature restoration with intra-block or inter-block feature restoration. Figure 7 shows the results of combining basic feature restoration with inter-block feature restoration. In Figure 7, the first convolutional layer of the current block was fed into the first convolutional layer of the next block, forming a connection as “skip”. Here, we give an abbreviated “1st-skip” to express that the first and second blocks are connected (i.e., the first convolutional layer of the first block was fed into the first convolutional layer of the second block), abbreviated “2nd-skip” to express that the second and third block are connected, and so forth. Meanwhile, “full-skip” represents all blocks that are connected in sequence from the first block to the last ReLU layer of the entire network. From Figure 7, we can see that the recognition accuracies of Baseline-20-basic and Baseline-36-basic are higher than Baseline-20-inter and Baseline-36-inter. This indicates that supplying inter-block features actually weakens the performance of the network with basic feature restoration.

In Figure 8, three blocks of Baseline-20 and Baseline-36 were fed into intra-block features. This shows that the recognition accuracys of Baseline-20 and Baseline-36 with only basic feature restoration are higher than those with both basic and intra-block feature restoration. This suggested that supplying intra-block features also weaken the network’s performance with basic feature restoration.

Thus, the above experimental results further demonstrated that the basic feature restoration strategy is optimal. Both inter-block and intra-block feature restoration strategies supplied excess features from the high network layer, while high layers always lose more information, and the deeper the layers, the more information is lost after the binarization of the full-precision model. It also brings in serious activation saturation, and the accuracy decreases.

Therefore, the proposed BinVPR will select the basic feature restoration strategy, and BinVPR’s network structure can be constructed. BinVPR-20 can be established based on Baseline-20, as shown in Figure 9. The basic feature restoration is performed on the last ReLU and the last convolutional layer of the network, achieving the highest accuracy. Similarly, BinVPR-36 can be established based on Baseline-36 as shown in Figure 10. Basic feature restoration is performed on the last ReLU and the last 1–8 convolutional layers, achieving the highest accuracy.

Although the differences in network configurations and datasets may result in varied structures of BNNs, our experiment found key principles for designing effective binary network architectures to address the gradient-vanishing problem encountered during training. The key principles are as follows:(1)It is essential to restore basic features to address gradient vanishing. The phenomenon of gradient vanishing cannot be avoided by restoring the binarization of inter-block and intra-block features because these features lose too much information after binarization.(2)The gradient-vanishing problem appeared during the training phase, caused by the loss of information in the high level of BNNs. Firstly, Our approach restored basic features to the top layer of the network and then gradually restored the features to the lower layer until this problem was solved. The structure of the network was determined.

### 4.4. Optimization for the Combination of Activations and Weights Function

Taking a binarization ResNet-18/34 with a full-precision short-cut connection (abbreviated as “BiResNet-18” and “BiResNet-34”) as an example, this work performed a randomized pairing procedure under Larq framework without separating the input activation and weight signs in BiResNet-18/34. In this section, experiments were conducted on “Forest”, 48 potential combinations as described in Section 3.4 were tested and results are given in Table 4 and Table 5.

The six rows in Table 4 and Table 5 give weight functions including Approx, STE, STE Tern, Swish, DoReFaNet and Magnitude Aware signs, while eight columns in tables give activation functions including Approx, STE, STE Tern, Swish, DoReFaNet, Magnitude Aware, Hard Tanh, and Leaky Tanh signs. We can see that the combination of “leaky tanh” as as an activation function and “DoReFa-Net” as as a weight function achieves the best accuracies for both ResNet-18 and ResNet-34, i.e., 77% and 75.75%. In terms of eight activation functions, we computed average accuracies for each one, results show that “leaky tanh” (with accuracy 63.04%) and “hard tanh” (with accuracy 63.54%) obtained the best performance for BiResNet-18, while “leaky tanh” [8] (with accuracy 63.54%) and “hard tanh” [8] (with accuracy 63.05%) achieved better accuracies for BiResNet-34. In contrast, “Magnitude Aware” is unsuitable for binarizing input activations in BiResNet-18/34 because it obtained “N/A” at all of rows, i.e., gradient vanishing emerges when combined with each weight function. Meanwhile, for BiResNet-34, more “N/A”s appear in Table 2, e.g., “Approx” also recieved “N/A” at all of rows, “Swish” signs recieved 5 “N/A”s. With respect to weight functions, both “Approx sign” (with accuracy 63.22%) and “DoReFa-Net”(with accuracy 63.41%) achieved better performance for BiResNet-18/34. It is worth noting that with all columns got “N/A”, “STE Tern” is cannot to quantify the weights of BiResNet-18/34. Overall, with the network growing deeper, the more difficult to handle the gradient-vanishing problem due to it losing more information.

Thus, “leaky tanh” is selected as the activation function, and “DoReFa-Net” is selected as the weight function in BinVPR as the optimal combination.

### 4.5. Comparison to State-of-the-Art Neural Networks

In this section, experiments were conducted on “Forest”, “Natural scene”, “Indoor scene”, and “Outdoor human-made building”. Our BinVPR-20 and BinVPR-36 were constructed, while ”BinVPR-STE-20/36” denotes that the STE sign binarizes input activations and weights in BinVPR-20/36. BinVPR-STE-20/36 was constructed because the STE sign represents the average performance of the aforementioned activation functions and weight functions in the last section. “BinVPR-Leaky-20/36” used the optimal combination that leaky tanh works as an activation function and DoReFaNet works as a weight function. They are compared with state-of-the-art binary neural networks (i.e., BinaryNet, XNOR-Net, FloppyNet, ShollowNet, DoReFaNet, BiRealNet-34, and RealToBinNet-34) and full-precision networks (i.e., AlexNet, ResNet-18 and ResNet-34). BiResNet-18 and BiResNet-34 were also compared. Results are given in Table 6, at a column of bitwidth, with 1-bit or 32-bit indicating that it is a binary or full-precision network. The performance was compared in terms of parameters, model size, and recognition accuracy.

From Table 6, we can see that the model size of all binary networks is less than 10 MB; in contrast, the model size of all full-precision networks is more than 40 MB. In particular, with 1.98 MB, the model size of BinVPR-20 is only 1% of AlexNet and 4.6% of ResNet-18. Thus, binarizing networks can greatly reduce the model size. Meanwhile, the number of parameters shows consistent results with model size.

In terms of recognition accuracy, the proposed BinVPR-20/36 surpasses all of the other binary networks on the four-class datasets and achieves performance comparable to full-precision networks. The performance of ResNet-34 is weaker than ResNet-18 in outdoor human-made scenes and forests, which is a real phenomenon. Compared with indoor and natural scenes, outdoor human-made buildings have a single repetitive structure and relatively simple image features. ResNet-34 will capture useless features, resulting in performance degradation. The forest dataset has fewer than ten scenarios. ResNet-34 may overfit in the training process, and its performance will be worse than ResNet-18. In particular, it can be seen that gradient-vanishing appears at XNOR-Net and RealToBinNet-34 on “Natural scene” and “Indoor scene”. While, “Natural scene” consists of natural images, “Indoor scene” focuses on indoor environments. Although these two datasets contain a wealth of information, the binarization process brings in much loss of information, that XNOR-Net and RealToBinNet-34 are not competent to solve the gradient-vanishing problem. In addition, the proposed BinVPR-20/36 is superior to AlexNet, and BinVPR-Leaky-20 presents the best performance on all datasets with accuracy of 78.25%, 53.85%, 52.16%, and 54.32%. BinVPR-Leaky-36 has a slightly lower accuracy than BinVPR-Leaky-20, but it surpasses ResNet-34 in terms of accuracy.

In summary, binarizing neural networks can significantly reduce the model size to meet the requirements of mobile robots. Moreover, by supplying basic features to high layers of the network, and combining “leaky tanh” as the activation function and “DoReFa-Net” as the weight function, the proposed BinVPR model achieves significant improvement in recognition accuracy and gradient vanishing problem also disappears.

## 5. Conclusions

This work proposed a new BinVPR model for BNN-based VPR methods to solve the gradient-vanishing problem that appeared in the training process and handle a marked drop in accuracy. For the gradient-vanishing problem, three feature restoration strategies were explored to add the lost information into higher layers, i.e., basic feature restoration, inter-block feature restoration, and intra-block feature restoration. The experimental results show that basic feature restoration is able to solve the gradient-vanishing problem. Meanwhile, the deeper the whole network, the more information is lost after the binarization of the full-precision model; thus, higher convolutional layers are needed to restore basic features. Furthermore, we have identified two principles for designing a structure of BNNs to address the problem of gradient vanishing: restoring basic features and restoring basic features from higher layers to lower layers in turn. To improve the dropped accuracy, a brute-force approach was used to find the optimal combination of binarized activation and binarized weight function in the Larq framework. Then, “leaky tanh” was selected as an activation function and “DoReFa-Net” was selected as a weight function in BinVPR as the optimal combination.

Finally, a baseline network based on ResNet was constructed, and the proposed BinVPR was established to realize visual place recognition. The performance of BinVPR was tested on public datasets. It was compared with state-of-the-art binary neural networks and full-precision networks (i.e., AlexNet, ResNet-18, and ResNet-34) in terms of parameters, model size, and recognition accuracy. Results show that BinVPR outperforms state-of-the-art BNN-based approaches and achieves the same accuracy with only 1% and 4.6% model size of AlexNet and ResNet. Further work will focus on exploring the complementary efficiency features in deeper networks and binarizing the VLAD layer in the VPR community.

## Figures and Tables

**Figure 1 sensors-24-04130-f001:**
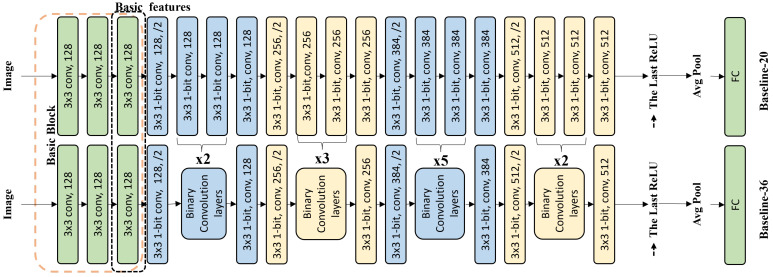
The baseline network: Baseline-20/36. For Baseline-36, “binary convolution layers” represents a few binarized convolutional layers; “xn” represents that it is constructed by cutting n corresponding binarized convolutional layers in Baseline-18, e.g., in Block 1; “x2” represents that “binary convolution layers” is established by cutting 2 corresponding binarized convolutional layers in Baseline-18, and the number of binary convolutional layers is 4; “x3” means the number of binary convolutional layers is 6, etc.

**Figure 2 sensors-24-04130-f002:**
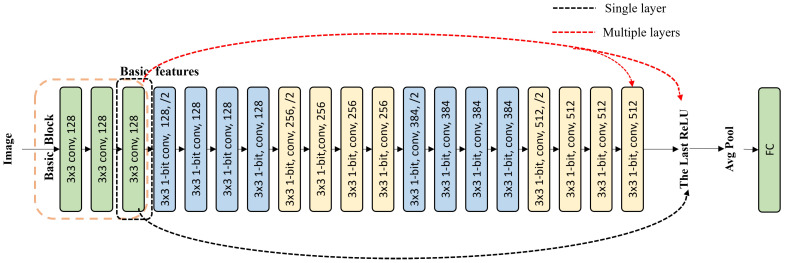
The strategy of basic feature restoration. In the restoration for a single layer way, basic features were added to a single layer. Here, we just give a case when basic features were added to the last ReLU. In the restoration for multiple layers way, basic features were supplied to multiple layers at the same time. Here, we just give a case when basic features were added to the last convolutional layer and the last ReLU.

**Figure 3 sensors-24-04130-f003:**
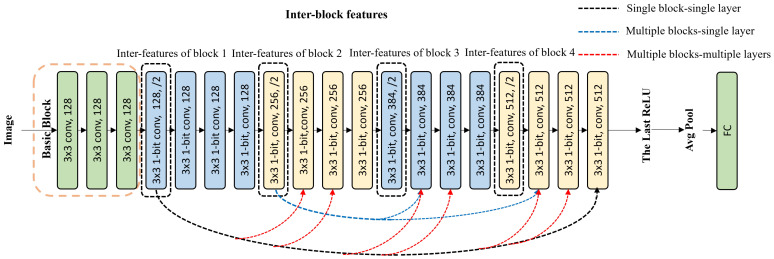
The strategy of inter-block feature restoration. For “single block - single layer”, the black dotted line shows a case where the inter-block features of Block 1 were added to the last layer of Block 4. For “multiple blocks-single layers”, the blue dotted line shows a case that the inter-block features of Block 2 were added to the second layer of Block 3 and the second layer of Block 4. For “multiple blocks - multiple layers”, the red dotted line shows a case that the inter-block features of Block 1 were added to the second and third layers of Blocks 2, 3, and 4, thus 6 convolutional layers accepted the restoration.

**Figure 4 sensors-24-04130-f004:**
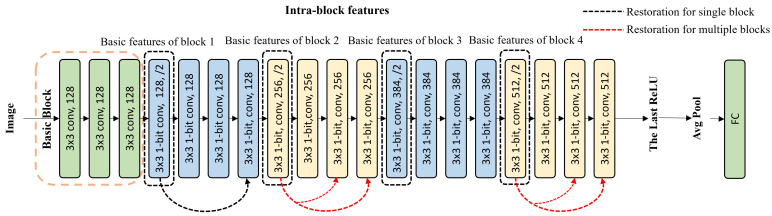
The strategy of intra-block feature restoration. The black dotted line shows a case where the basic features of Block 1 were added to the last convolutional layer of Block 1. The red dotted line gives a case where the basic features of Block 2 were added to the third and last convolutional layers; meanwhile, the basic features of Block 4 were added to the third and last convolutional layers.

**Figure 5 sensors-24-04130-f005:**
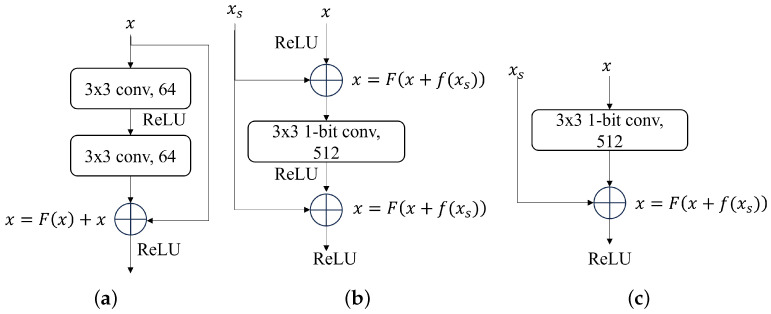
The original shortcut connection in ResNet18/34 and ours method: supply features for ReLU and convolutional layer. (**a**) Origin Shortcut. (**b**) Supply features in convolutional layer. (**c**) Supply features in ReLU.

**Figure 6 sensors-24-04130-f006:**
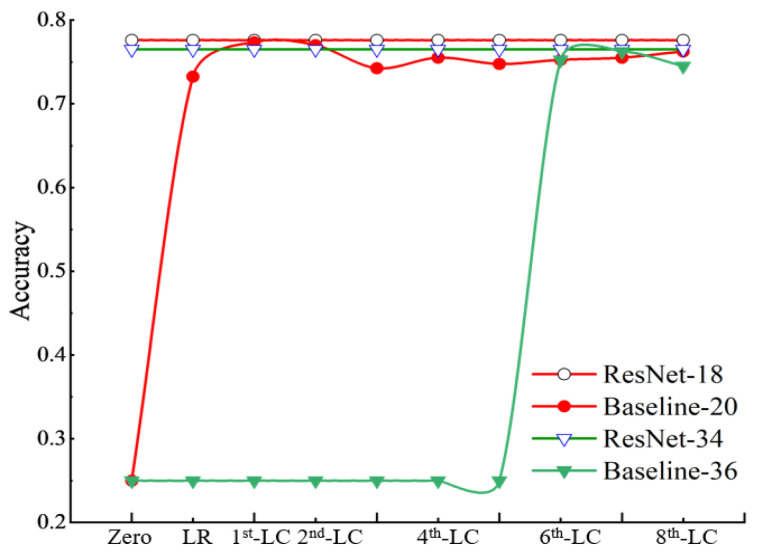
Results of basic feature restoration for multiple layers. The *y*-axis shows accuracy, and the *x*-axis gives places where basic features were restored.

**Figure 7 sensors-24-04130-f007:**
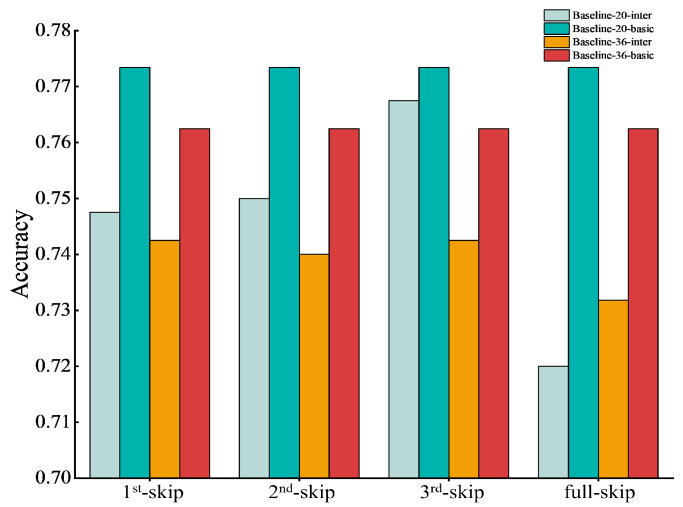
Results of combinating basic features and Inter-block features restoration. The *y*-axis shows recognition accuracy, and the *x*-axis presents cases where the block was connected in inter-block feature restoration. Baseline-20/36-basic represents the results when only basic features were added. Baseline-20/36-inter represents the results when basic features and inter-block feature restorations were combined.

**Figure 8 sensors-24-04130-f008:**
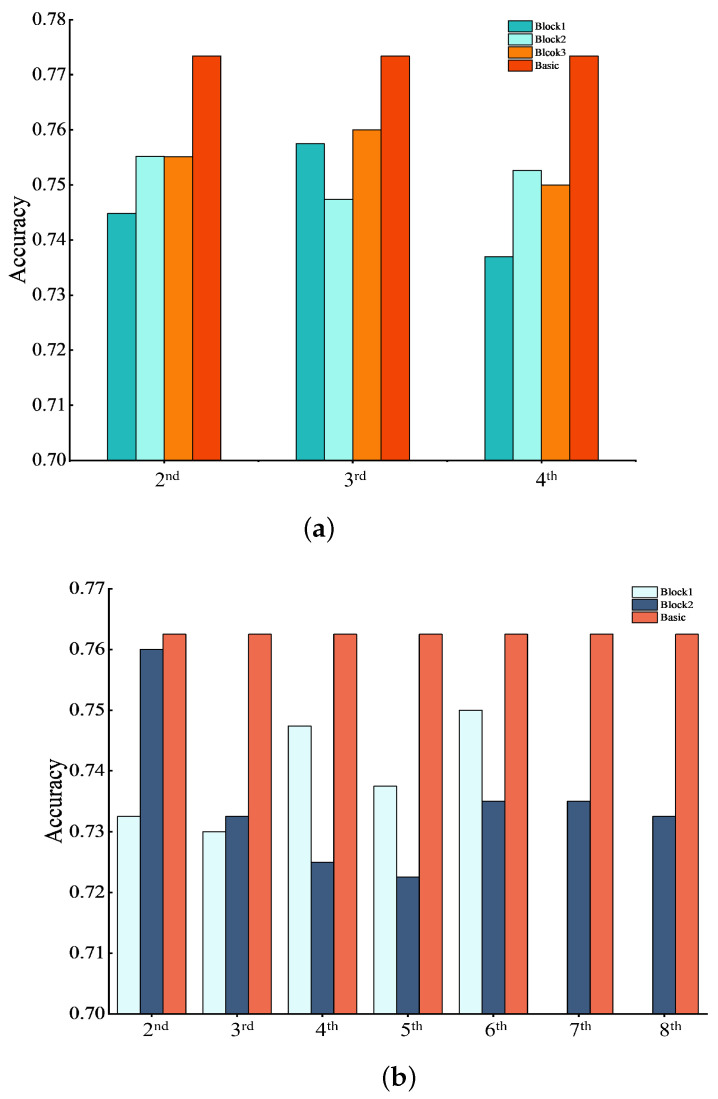
Results of combinating basic features and intra-block features. *y*-axis shows recognition accuracy, and the *x*-axis represents the layers that accept the first convolutional layer in each block. “Basic” represents only basic features were added to Baseline-20 and Baseline-36. (**a**) Combination of basic features and intra-block in Baseline-20. (**b**) Combination of basic features and intra-block features in Block 1 and Block 2 of Baseline-36. (**c**) Combination of basic features and intra-block features in Block 3 of Baseline-36.

**Figure 9 sensors-24-04130-f009:**
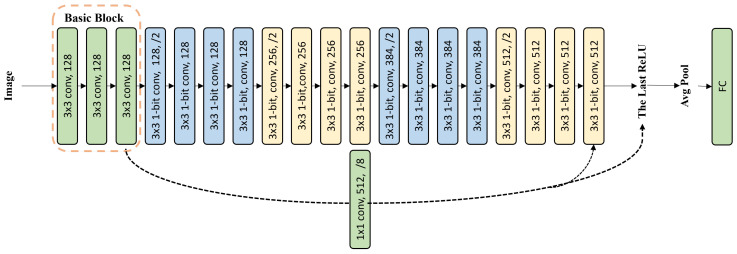
The network architecture of BinVPR-20.

**Figure 10 sensors-24-04130-f010:**
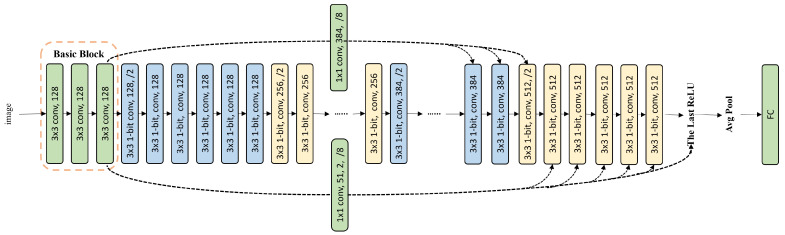
The network architecture of BinVPR-36.

**Table 1 sensors-24-04130-t001:** Summary of datasets for evaluation.

	Structural Information	Environment			Variation			
		Urban	Suburban	Natural	Viewpoint	Dynamic	B	Weather
Forests	−			✔	++	++	++	++
Natural	−		✔	✔	+	+	++	+
Outdoor	+	✔	✔	✔	+	+	+	+
Indoor	++	✔			+	−	−	−

++, +, and − indicates the degree of environmental change from high to low. “Natural” represents “Natural scene”. “Outdoor” represents “Outdoor human-man building”. “Indoor” represents “Indoor scene”.

**Table 2 sensors-24-04130-t002:** Restore inter-block feature in Baseline-20.

	Convolutional Layer	1st B1	1st B2	1st B3
	2nd	N/A	−	−
Block2	3rd	N/A	−	−
	4th	N/A	−	−
	2nd	N/A	N/A	
Block3	3rd	N/A	N/A	−
	4th	N/A	N/A	−
	2nd	N/A	N/A	N/A
Block4	3rd	N/A	N/A	N/A
	4th	N/A	N/A	N/A
	2nd	N/A	−	−
Block2+3	3rd	N/A	−	−
	4th	N/A	−	−
	2nd	N/A	N/A	−
Block3+4	3rd	N/A	N/A	−
	4th	N/A	N/A	−
	2nd	N/A	−	−
Block2+3+4	3rd	N/A	−	−
	4th	N/A	−	−
	2nd + 3rd	N/A	−	−
Block2+3	3rd + 4th	N/A	−	−
	2nd + 3rd + 4th	N/A	−	−
	2nd + 3rd	N/A	N/A	−
Block3+4	3rd + 4th	N/A	N/A	−
	2nd + 3rd + 4th	N/A	N/A	−
	2nd + 3rd layer	N/A	−	−
Block2+3+4	3rd + 4th	N/A	−	−
	2nd + 3rd + 4th	N/A	−	−

“B” represents Block. “1st B1” represents the first layer of block1. “1st B2” and “1st B3” have similar meanings. “1st” represents the first layer, and “2nd”, “3rd”, and “4th” have similar meanings. “‘−” represents there is no experimentation. “N/A” represents that gradient vanishing appears.

**Table 3 sensors-24-04130-t003:** Restore intra-block feature in Baseline-20.

	The Same Block		
	2nd	2nd + 3rd	2nd + 3rd + 4st
1st layer of B1	N/A	N/A	N/A
1st layer of B2	N/A	N/A	N/A
1st layer of B3	N/A	N/A	N/A
1st layer of B4	N/A	N/A	N/A
1st layer of B1+2	N/A	N/A	N/A
1st layer of B1+2+3	N/A	N/A	N/A
1st layer of B1+2+3+4	N/A	N/A	N/A

“B” represents Block. “N/A” represents that gradient vanishing appears. “2nd” represents the second layer. “2nd+3rd” represent the second and third layers. “2nd+3rd+4st” has a similar meaning. “1st layer of B1” represents the first layer of Block 1. The second through fourth rows have a similar meaning. “1st layer of B1+2” represents the first layers of Block 1 and Block 2. row 6 and 7 have a similar meaning.

**Table 4 sensors-24-04130-t004:** Accuracies from different combinations of activations and weights sign in BiResNet-18.

	Activation	Approx Sign	STE Sign	STE Tern	Swish Sign	DoReFa Activ	Magn Sign	Hard Tanh	Leaky Tanh	Average Accuracy
Weight										
Approx Sign [27]		68.50%	68.50%	74.25%	67.50%	75.50%	N/A	74.50%	**77.00%**	63.22%
STE Sign [24]		67.50%	69.50%	74.50%	68.75%	74.25%	N/A	75.75%	75.50%	63.22%
STE Tern [40]		N/A	N/A	N/A	N/A	N/A	N/A	N/A	N/A	0%
Swish Sign [41]		66.75%	71.50%	76.25%	65.00%	74.75%	N/A	75.25%	76.00%	63.19%
DoReFa Wei [26]		69.50%	67.00%	75.50%	67.00%	74.75%	N/A	76.50%	**77.00%**	63.41%
Magn [27]		67.25%	67.75%	74.00%	69.00%	75.00%	N/A	76.25%	75.75%	63.06%
Average Accuracy		56.58%	57.38%	62.42%	56.21%	62.38%	0%	63.05%	63.54%	

“N/A” represents gradient vanishing appearing. “Activation” represents binarization activation function. “DoReFa Wei” represents DoReFaNet’s weight function. “DoReFa Activ” represents DoReFaNet’s activation function. ”Magn” represents ”Magnitude Aware Sign”. The bold data in the table is the best combination of binarized functions.

**Table 5 sensors-24-04130-t005:** Accuracies from different combinations of activation and weight signs in BiResNet-34.

	Activation	Approx Sign	STE Sign	STE Tern	Swish Sign	DoReFa Activ	Magn Sign	Hard Tanh	Leaky Tanh	Average Accuracy
Weight										
Approx Sign		N/A	67.75%	**76.25%**	N/A	75.50%	N/A	72.25%	74.50%	45.78%
STE Sign		N/A	67.00%	73.25%	N/A	73.50%	N/A	74.50%	73.66%	45.24%
STE Tern		N/A	N/A	N/A	N/A	N/A	N/A	N/A	N/A	0%
Swish Sign		N/A	66.25%	73.75%	N/A	74.00%	N/A	72.25%	75.50%	45.22%
DoReFa wei		N/A	62.50%	74.25%	74.25%	72.25%	N/A	75.00%	**75.75%**	54.25%
Magn		N/A	N/A	74.00%	N/A	72.00%	N/A	71.98%	74.75%	36.59%
Average Accuracy		0%	43.25%	61.92%	12.38%	61.21%	0%	61.50%	62.36%	

“N/A” represents gradient vanishing appearing. “Activation” represents binarization activation function. “DoReFa Wei” represents DoReFaNet’s weight function. “DoReFa Activ” represents DoReFaNet’s activation function. ”Magn” represents ”Magnitude Aware Sign”. The bold data in the table is the best combination of binarized functions.

**Table 6 sensors-24-04130-t006:** Comparison with state-of-the-art networks.

Neural Network	Bitwidth [bit]	Parameters [Million]	Model Size [MiB]	Forest Accuracy	Natural Accuracy	Indoor Accuracy	Outdoor Accuracy
BinaryNet	1	71.71	8.55	73.70%	39.11%	37.24%	36.47%
XNOR-Net	1	66.00	7.87	74.48%	N/A	N/A	46.69%
FloppyNet	1	3.68	0.44	73.18%	44.12%	36.17%	39.64%
ShallowNet	1	55.98	6.67	73.44%	43.54%	38.60%	40.15%
DoReFaNet	1	72.13	8.60	70.25%	46.32%	44.13%	45.71%
RealToBinNet-34	1	21.89	2.61	68.34%	N/A	N/A	47.81%
BiRealNet-34	1	21.35	2.55	73.18%	39.80%	23.93%	24.24%
BiResNet-18	1	11.28	1.34	53.91%	51.96%	17.05%	16.87%
BiResNet-34	1	21.32	2.54	58.75%	51.22%	16.43%	18.24%
AlexNet	32	58.50	223.18	64.84%	49.51%	46.23%	47.82%
ResNet-18	32	11.28	42.76	77.75%	52.49%	46.60%	53.69%
ResNet-34	32	21.32	81.35	76.75%	52.98%	51.01%	52.81%
BinVPR-STE-20	1	16.63	1.98	77.50%	53.65%	38.01%	53.16%
BinVPR-STE-36	1	34.69	4.14	76.50%	50.11%	44.40%	53.03%
BinVPR-Leaky-20	1	16.63	**1.98**	**78.25%**	**53.85%**	**52.16%**	**54.32%**
BinVPR-Leaky-36	1	34.69	4.14	77.29%	52.47%	51.17%	52.35%

“N/A” represents gradient vanishing appearing. “BinVPR-STE-20” means that the STE sign is used to binarize input activations and weights in BinVPR-20. “BinVPR-Leaky-20” means that the Leaky Tanh sign is used to binarize input activations and the DoReFaNet weight sign is used to binarize weights in BinVPR-20. “BinVPR-STE-36” and “BinVPR-Leaky-36” are similar to the above. ”Outdoor” represents ”Outdoor human-made building”. ”Natural” represents ”Natural scene”. ”Indoor” represents ”Indoor scene”.

## Data Availability

The datasets used and analyzed in the current study are available from the corresponding author on reasonable request. The source code is publicly available at: https://github.com/Junshuai1043/BinVPR (accessed on 6 May 2024).

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
