# Peer review of "BinVPR: Binary Neural Networks towards Real-Valued for Visual Place Recognition"

_sensors, 2024, doi:10.3390/s24134130_

Round 1

Reviewer 1 Report

Comments and Suggestions for Authors

This papers mainly concentrates on addressing gradients vanishing and accuracy drop problems in Binary Neural Networks (BNNs) based Visual Place Recognition (VPR). The authors do plenty of experiments testing and analyzing different methods in order to design a new optimal framework. However, there are still some details and questions should be considered:

1. Popular VPR methods are compared by top1 accuracy and top5 accuracy while there is only one accuracy in the article. It would be better to have an explanation on which one this accuracy is and whether this is sufficient.

2. The experiments were conducted on a part of Place365. Is the chosen part fair to compare?  If this would affect the final results?  It would be better to have some explanations about this in the paper.

3. As the authors show in Table6, full-precision ResNet-34 has weaker accuracy comparing to full-precision ResNet-18.  However, ResNet-34 has more powerful ability of representing than ResNet-18. So whether this is concerned with inadequate training and data supplements or it is just a real phenomenon in experiments?  

4. Some writing errors should be carefully checked. For example, two capital letters appear in one sentence in line 175.

5. There are many grammar errors and improper statements which influence understanding. For example, it would be better to replace ‘How features are restored’ in line 344 and ‘What features are restored?’ in line 258 with normal questions

Comments on the Quality of English Language

The language should be improved.

Author Response

Authors' Response to Reviewers Comments

Paper ID: sensors-2953259

First of all, the authors would like to thank the Editor, Associate Editor, and Reviewers for their careful examination and constructive comments on the manuscript titled “BinVPR: Binary Neural Networks Towards Real-Valued for Visual Place Recognition”, which have helped improving the presentation and quality of the paper significantly. In the revision, the authors have addressed all of Reviewers’ comments, and have made the corresponding modifications on the paper in details. The modifications with respect to the previous version have been indicated in blue. The explanations of revisions and point-to-point responses to the comments are provided as follows. We hope that this version can be satisfactory for acceptance.

We feel great thanks for your professional review work on our article. According to your nice suggestions, we have made extensive corrections to our previous manuscript, the detailed corrections are listed below.

Comment 1: Popular VPR methods are compared by top1 accuracy and top5 accuracy while there is only one accuracy in the article. It would be better to have an explanation on which one this accuracy is and whether this is sufficient.

Reply:

We used Top-1 accuracy in the paper. Top-1 accuracy means the best guess (class with highest probability) is the correct result. Top-5 accuracy means the correct result is in the top 5 best guesses (5 classes with highest probabilities). Therefore, compared with Top-5 accuracy, Top-1 accuracy is more serious and straightforward, as it measures the model's ability to predict the most likely class correctly. Moreover, in some of the datasets we trained, the maximum categories are 10, Top-5 accuracy cannot provide enough discrimination, and most of the predictions locate in the top five. Therefore, we chose Top-1 accuracy as the evaluation metric. The explanation is added and highlighted in blue on lines 376 to 383 of the paper.

Comment 2: The experiments were conducted on a part of Place365. Is the chosen part fair to compare? If this would affect the final results? It would be better to have some explanations about this in the paper.

Reply:

Thanks for the reviewer's professional advice. According to the reviewer's suggestions, we added clearly explanations about the datasets used in the paper.

Firstly, all images of Places365 were used in the experiment. We divided the Places365 dataset into four categories: Forest, Natural scene, Indoor scene, and Outdoor human-made scene. VPR faces the challenge of perceptual aliasing and variability in changing environment. Due to seasonal changes, lighting conditions, and object occlusion, the place's appearance changes greatly, leading to perceptual aliasing. In addition, when the structure in a  scene looks similar, it is easy to cause perceptual variability. The forest environment is complex, with all of above challenges that it is  constructed as a category “Forest”. Torii[1] divided scene into two categories based on whether or not the image has a repetitive structure: human-made scene and natural scene. Based on [1], this work divided human-made scene  further into indoor human-made scene and outdoor human-made scene based on whether or not the appearance of scene was greatly affected by natural factors such as light and seasonal changes. In contrast, indoor scene, VPR is usually unaffected by light and seasonal changes. Finally, the structures of natural scene are similar but not repetitive and are also significantly affected by natural factors. Thus, we subdivided the Places365 dataset into four categories to improve the accuracy of VPR and used all data to test and verify BNNs in experiments. It does not affect the validity of experimental results. The explanation is highlighted in blue on lines 348 to 367 of the paper.

[1] A. Torii, J. Sivic, M. Okutomi and T. Pajdla, "Visual Place Recognition with Repetitive Structures," in IEEE Transactions on Pattern Analysis and Machine Intelligence, vol. 37, no. 11, pp. 2346-2359, 1 Nov. 2015, doi: 10.1109/TPAMI.2015.2409868.

Comment 3: As the authors show in Table6, full-precision ResNet-34 has weaker accuracy comparing to full-precision ResNet-18. However, ResNet-34 has more powerful ability of representing than ResNet-18. So whether this is concerned with inadequate training and data supplements or it is just a real phenomenon in experiments?  

Reply:

The performance of ResNet-34 is weaker than ResNet-18 in outdoor human-made scene and forest, which is a real phenomenon. There are many types of data sets in outdoor environments. Compared with indoor and natural scene, outdoor human-made building has a single repetitive structure and relatively simple image features. ResNet-34 will capture useless features, resulting in performance degradation. The “Forest” has fewer than ten scenarios. ResNet-34 may overfit in the training process, and its performance will be worse than ResNet-18. The modified parts are highlighted in blue on lines 547 to 553.

Comment 4: Some writing errors should be carefully checked. For example, two capital letters appear in one sentence in line 175.

Reply:

We sincerely thank the reviewer for careful reading. As Reviewer suggested that we have corrected the error in line 175. At the same time, we also made corresponding checks in other parts of the article. The modified parts are highlighted in blue.

Comment 5: There are many grammar errors and improper statements which influence understanding. For example, it would be better to replace ‘How features are restored’ in line 344 and ‘What features are restored?’ in line 258 with normal questions.

Reply:

Thank for reviewer positive comments and valuable suggestions to improve the quality of our manuscript. We used ‘How features are restored?’ to replace ‘What features are restored?’ and used ‘What features are restored?’ to replace ‘Where to restore features?’. We have also made corresponding changes in other parts of the manuscript. The modified parts are highlighted in blue.

Reviewer 2 Report

Comments and Suggestions for Authors

Pros:

 * Paper proposes a potentially parameter-efficient convolutional neural network (CNN) binarization approach for visual place recognition (VPR), and conducts its experimental evaluation. Empirical results reveal that the proposed model tends to be more efficient than other investigated counterparts. However, this study also seems to have some serious drawbacks as well.

Cons:

* English should be carefully checked and corrected.

* I would recommend augmenting the Related Work (Section 2) with the discussion about potential applications of VPR (e.g. visual SLAM, Pre-Training of neural autonomous driving agents, etc.)

* Some formulas are with errors (e.g. not clear how to interepret maximum of one value (probably maximum is selected by all indices (?) in (6), (8) is with typo, etc.)

* No code repository provided. In my opinion, authors should either include a code repository, or at least polish and clarify their algorithmic contributions (e.g. at least in pseudocode) to make their study (at least more) replicable.

* The data set should be described more clearly. It is unclear whether it is composed as a subset of Places365 (see text in line 303). Why this single data set was selected for the experiments?

* Some recent results in binarization [1] and VPR [2,3] may also be included in the bibliography.

[1] https://arxiv.org/abs/2310.11453

[2] https://www.sciencedirect.com/science/article/abs/pii/S0925231222012188

Comments on the Quality of English Language

* English should be carefully checked and corrected.

Author Response

Authors' Response to Reviewers Comments

Paper ID: sensors-2953259

First of all, the authors would like to thank the Editor, Associate Editor, and Reviewers for their careful examination and constructive comments on the manuscript titled “BinVPR: Binary Neural Networks Towards Real-Valued for Visual Place Recognition”, which have helped improving the presentation and quality of the paper significantly. In the revision, the authors have addressed all of Reviewers’ comments, and have made the corresponding modifications on the paper in details. The modifications with respect to the previous version have been indicated in blue. The explanations of revisions and point-to-point responses to the comments are provided as follows. We hope that this version can be satisfactory for acceptance.

We feel great thanks for your professional review work on our article. According to your nice suggestions, we have made extensive corrections to our previous manuscript, the detailed corrections are listed below.

Comment 1:  English should be carefully checked and corrected.

Reply:

Thanks for the reviewer's professional advice. For the language, we invite a native English speaker to help polish all of our articles to make them more academic, fluid, and easier to understand. Furthermore, we carefully checked spelling, grammar, and format, corrected errors, and adjusted formatting. We hope the revised manuscript will be acceptable.

Comment 2: I would recommend augmenting the Related Work (Section 2) with the discussion about potential applications of VPR (e.g. visual SLAM, Pre-Training of neural autonomous driving agents, etc.)

Reply:

Thanks for the reviewer's professional advice. According to the reviewer's comments, we added the discussion about potential applications of VPR into Related Work part as follows:

VSLAM (Visual Simultaneous Localization and Mapping) enables mobile robots to map an unknown environment and determine its location simultaneously. It comprises visual odometer, loop closure detection (LCD), back-end optimization, and mapping components. True loop closure reduces the cumulative position errors caused by visual odometers and builds accurate and consistent maps. The LCD, also regarded as place recognition, verifies if an image location exists in the environment database. VPR (Visual Place Recognition) in vSLAM employs local feature descriptors (e.g., SIFT, SURF, ORB), global feature descriptors (e.g., WI-SURF, BRIEF-Gist), and learning methods. Local descriptors are robust to rotation, scaling, and partial occlusion, while global descriptors are resilient to illumination changes but weaker to occlusion and lack of geometric information. Learning-based VPR offers advantages of both but demands high computing and storage, making resource reduction a key research focus. VPR identifies key elements such as road markings, traffic signs, vehicles, and pedestrians from the image through feature extraction and matching algorithms. This method can provide the vehicle's location information and information about the vehicle's environment, which provides essential support for the decision planning and control execution of the autonomous driving system. The addition parts are highlighted in blue on lines 154 to 189.

Comment 3: Some formulas are with errors (e.g. not clear how to interepret maximum of one value (probably maximum is selected by all indices (?) in (6), (8) is with typo, etc.)

Reply:

We sincerely thank the reviewer for careful reading. The maximum value is taken over all weights in this layer in (6). We corrected the spelling of formula (8) by missing a parenthesis. In the meantime, we also checked other formulas in the article.

Comment 4: No code repository provided. In my opinion, authors should either include a code repository, or at least polish and clarify their algorithmic contributions (e.g. at least in pseudocode) to make their study (at least more) replicable.

Reply: As Reviewer suggested that we upload our code to the following link:  https://github.com/Junshuai1043/BinVPR.

Comment 5:  The data set should be described more clearly. It is unclear whether it is composed as a subset of Places365 (see text in line 303). Why this single data set was selected for the experiments?

Reply:

Thanks for the reviewer's professional advice. According to the reviewer's suggestions, we added clearly explanations about datasets this in the paper.

Firstly, all images of Places365 were used in the experiment. We divided the Places365 dataset into four categories: Forest, natural scene, indoor scene, and outdoor human-made scene. VPR faces the challenge of perceptual aliasing and variability in changing environment. Due to seasonal changes, lighting conditions, and object occlusion, the place's appearance changes greatly, leading to perceptual aliasing. In addition, when the location structure of the scene is similar, it is easy to cause perceptual variability. The forest environment is complex, and the above challenges exist, so they are divided into one category. Inspired by the work of Torii[1], we divided scene into two categories based on whether or not the image has a repetitive structure: human-made scene and natural scene. While in this work, human-made scene are divided into indoor scene and outdoor human-made scene. Due to the outdoor human-made scene has many repetitive structures and was also greatly affected by natural factors such as light and seasonal changes. In contrast, in indoor scene, VPR is usually unaffected by light and seasonal changes, but has a large number of repetitive structures. Finally, the structures of natural scene are similar but not repetitive and are also  affected by natural   factors. Thus, we subdivided the Places365 dataset into four categories to improve the accuracy of VPR and used all data to test and verify BNNs in experiments. It does not affect the validity of experimental results. Therefore, recognizing the location in forest scene is more challenging than in other scenes. BNNs work well in forest scene and in other types of scenarios as well.

We chose the Places365 dataset as the experimental dataset. Places365 conforms to human visual cognition and can be used to train artificial neural networks for advanced visual tasks. It contains over a million images and has 365 image categories, each with 3,000 to 5,000 images. On the other hand, VPR can be divided into the four types of scenes mentioned above, and fewer datasets have four types of scenes simultaneously. Common datasets for VPR include Nordland, MSLS, Tokyo24/7, GSV-Cites, Oxford RobotCar, etc. Nordland and Oxford RobotCar collected vehicle driving data, including outdoor human-made and natural scene. GSV-Cites, Tokyo24/7, and MSLS collected urban street data, that is, outdoor man-made scene. Places365 contains four types of scenarios, which is more comprehensive than the Nordland, MSLS and other datasets. Places365 is suitable for testing the ability of neural networks to recognize locations, so we chose Places365. The explanations are highlighted in blue on lines 348 to 367 of the paper.

[1] A. Torii, J. Sivic, M. Okutomi and T. Pajdla, "Visual Place Recognition with Repetitive Structures," in IEEE Transactions on Pattern Analysis and Machine Intelligence, vol. 37, no. 11, pp. 2346-2359, 1 Nov. 2015, doi: 10.1109/TPAMI.2015.2409868.

Comment 6: Some recent results in binarization [1] and VPR [2,3] may also be included in the bibliography.

Reply:

Thanks for the reviewer's valuable comments, we added an introduction to recent literature [2,4,5]. We looked for recent papers on binarization [3] and VPR[6], and their papers are included in the bibliography.

Wang et al. [2]proposed BitNet for large language model, which is a scalable and stable 1-bit transformer architecture. They designed the trainable 1-bit fully connected layer BitLinear in place of nn.Linear layer. BitNet achieved competitive performance while reducing memory footprint and energy consumption, compared to full-precision Transformer baselines. The addition parts are highlighted in blue on lines 141 to 145.

Xu et al. [3] proposed ReBNN to improve learning ability. This method reduced the loss of binarization process by calculating the balanced parameter based on its maximum magnitude. The addition parts are highlighted in blue on lines 145 to 147.

Izquierdo et al. [4]proposed SALAD (Sinkhorn Algorithm for Locally Aggregated Descriptors). Considering the feature-to-cluster and cluster-to-feature relationships, they introduced 'dustbin' cluster to selectively discard the features that are considered to be non-informative and improve the overall quality of the descriptors. The addition parts are highlighted in blue on lines 109 to 113.

It is worth noting that Ali-bey proposed GSV-Cities[5] dataset to address the challenge of lacking large databases with accurate ground truth. Ali-bey collected more than 40 cities across all continents over a 14-year period, providing the widest geographic coverage and highly accurate on the ground truth. The addition parts are highlighted in blue on lines 113 to 117.

To realize seamless adaptation of the pre-trained transformer model for VPR, Lu et al. [6] proposed the SelaVPR model. This model design is a hybrid adaptation that adjusts the lightweight adapter. Besides, they proposed a mutual nearest neighbor local feature loss to guide effective adaptation, which reduces the matching time of image. The addition parts are highlighted in blue on lines 117 to 120.

[2] Wang, Hongyu et al. "BitNet: Scaling 1-bit Transformers for Large Language Models", CoRR abs/2310.11453 (2023)

[3] Xue, P., Lu, Y., Chang, J., Wei, X., & Wei, Z. (2023). Fast and Accurate Binary Neural Networks Based on Depth-Width Reshaping. AAAI Conference on Artificial Intelligence.

[4] Izquierdo, S., & Civera, J. (2023). Optimal Transport Aggregation for Visual Place Recognition. ArXiv, abs/2311.15937.

[5] Ali-bey, Amar et al. "GSV-Cities: Toward appropriate supervised visual place recognition", Neurocomputing 513 (2022): 194-203.

[6] Lu, Feng et al. "Towards Seamless Adaptation of Pre-trained Models for Visual Place  Recognition", International Conference on Learning Representations abs/2402.14505 (2024)

Round 2

Reviewer 1 Report

Comments and Suggestions for Authors

The authors have addressed all the comments.